# BUB1 Promotes Gemcitabine Resistance in Pancreatic Cancer Cells by Inhibiting Ferroptosis

**DOI:** 10.3390/cancers16081540

**Published:** 2024-04-18

**Authors:** Weiming Wang, Xiang Zhou, Lingming Kong, Zhenyan Pan, Gang Chen

**Affiliations:** 1Department of Hepato-Pancreatic Surgery, The First Affiliated Hospital of Wenzhou Medical University, Wenzhou 325035, China; wwm_boy2010@163.com (W.W.); klmtc2010@163.com (L.K.); pzy990907@163.com (Z.P.); 2Department of Breast Cancer, The First Affiliated Hospital, Wenzhou Medical University, Wenzhou 325035, China; zhouxiang@wzhospital.cn

**Keywords:** pancreatic cancer, ferroptosis, BUB1, gemcitabine, drug resistance

## Abstract

**Simple Summary:**

Further investigation into the molecular mechanisms of chemotherapy resistance in pancreatic cancer is crucial for improving the prognosis of pancreatic cancer patients. Extensive research has demonstrated a close association between ferroptosis and gemcitabine resistance in various tumor types, including pancreatic cancer. We first show that high expression of Mitotic checkpoint serine/threonine kinase BUB1 in pancreatic cancer and its association with poor prognosis. Results also suggested that BUB1 suppresses ferroptosis in pancreatic cancer cells, and BUB1 knockdown significantly enhances the sensitivity of drug-resistant pancreatic cancer cells to ferroptosis. Furthermore, BUB1 promotes gemcitabine resistance in pancreatic cancer cells. We believe that our study makes a significant contribution to the literature because it was previously unclear whether BUB1 plays a key role in pancreatic cancer ferroptosis.

**Abstract:**

The development of chemotherapy resistance severely limits the therapeutic efficacy of gemcitabine (GEM) in pancreatic cancer (PC), and the dysregulation of ferroptosis is a crucial factor in the development of chemotherapy resistance. BUB1 Mitotic Checkpoint Serine/Threonine Kinase (BUB1) is highly overexpressed in PC patients and is closely associated with patient prognosis. However, none of the literature reports the connection between BUB1 and ferroptosis. The molecular mechanisms underlying GEM resistance are also not well understood. Therefore, this study first established the high expression levels of BUB1 in PC patients, then explored the role of BUB1 in the process of ferroptosis, and finally investigated the mechanisms by which BUB1 regulates ferroptosis and contributes to GEM resistance in PC cells. In this study, downregulation of BUB1 enhanced the sensitivity of PC cells to Erastin, and inhibited cell proliferation and migration. Mechanistically, BUB1 could inhibit the expression levels of Neurofibromin 2 (NF2) and MOB kinase activator 1 (MOB1), and promote Yes-associated protein (YAP) expression, thereby inhibiting ferroptosis and promoting GEM resistance in PC cells. Furthermore, the combination of BUB1 inhibition with GEM exhibited a synergistic therapeutic effect. These findings reveal the mechanisms underlying the development of GEM chemotherapy resistance based on ferroptosis and suggest that the combined use of BUB1 inhibitors may be an effective approach to enhance GEM efficacy.

## 1. Introduction

Pancreatic cancer (PC) is a highly invasive solid tumor and ranks as the fourth leading cause of cancer-related deaths worldwide [1,2]. Among them, pancreatic ductal adenocarcinoma (PDAC) is the most common, aggressive, and lethal form of pancreatic cancer [1]. Furthermore, PC has an extremely poor prognosis, with a 5-year survival rate of only about 2–9% [1]. Currently, gemcitabine (GEM) remains a frontline chemotherapy drug for PC, primarily functioning by inducing cancer cells to undergo apoptosis and ferroptosis [3,4]. However, the development of resistance severely limits the effectiveness of GEM [5]. Therefore, further investigation into the molecular mechanisms of chemotherapy resistance in PC is crucial for improving the prognosis of PC patients.

Ferroptosis is a distinct form of cell death, different from apoptosis, necrosis, or autophagy, characterized by the accumulation of lipid peroxidation in the cellular membrane system, which is dependent on intracellular iron and reactive oxygen species (ROS) [6,7]. Ferroptosis plays a significant role in various pathological processes, including the development of chemotherapy resistance in several diseases [7]. The idea of utilizing ferroptosis in PC cells to address chemotherapy resistance has been proposed by Yang et al. [8], which served as inspiration for our study. The Hippo pathway, which mediates targeted ferroptosis, has been identified as a promising strategy in cancer treatment [9,10]. In recent years, increasing evidence suggests that the Hippo pathway may also play a crucial role in the process of chemotherapy resistance [11]. The Hippo signaling pathway consists of various regulatory factors, including Neurofibromin 2 (NF2), MOB kinase activator 1 (MOB1), and Yes-associated protein (YAP), and it regulates organ development and diseases. YAP is one of the major effector molecules of the Hippo pathway and is inhibited by the Hippo pathway [11,12,13]. Jiao Wu et al. reported that NF2-YAP signaling plays a critical role in determining the process of ferroptosis in cancer cells [9]. Ashley L. Hein et al. found that the PR55A regulatory subunit of PP2A can activate YAP by inhibiting the MOB1/LATS cascade in PC cells [14]. Furthermore, the YAP pathway has been identified as a major determinant of clinical invasiveness in PDAC patients, and the inhibition of YAP signaling is associated with a favorable prognosis [15]. Based on this, we speculate that the NF2/MOB1-YAP pathway may serve as a crucial regulatory pathway for ferroptosis.

Mitotic checkpoint serine/threonine kinase BUB1 is well-known for its regulatory function in the process of cell mitosis [16]. Aberrations in BUB1 can lead to chromosomal instability and promote tumor development [17,18]. BUB1 has been reported to be closely associated with the occurrence of various cancer types, including gastric cancer, breast cancer, and PDAC [19,20,21]. Furthermore, a recent study found that high expression of BUB1 is associated with poor overall survival in PDAC [16]. In a study investigating the molecular mechanisms of chemoresistance in bladder cancer cells, upregulation of BUB1 was identified as a significant factor in GEM resistance [22]. Another study revealed that the combination of FDI-6 and Olaparib inhibits PC growth by targeting the BUB1, BRCA1, and CDC25A signaling pathways [23]. Therefore, we consider that BUB1 may be involved in GEM resistance in PC cells, and the inhibition of BUB1 in combination with GEM treatment could significantly suppress PC development. Furthermore, upregulation of BUB1 and downregulation of YAP are often observed in the poor prognosis of PDAC [15,24]. In a leucine-deprived model of PC cells, downregulation of BUB1 and upregulation of NF2 protein have been observed [25]. Hence, we speculate that BUB1 may inhibit ferroptosis in PC cells and promote resistance to GEM by inhibiting the expression levels of NF2 and MOB1, leading to the activation of YAP.

In this study, we first compared the expression levels of BUB1 between PC patients and the normal population. Furthermore, we investigated the role of BUB1 in the process of ferroptosis. Subsequently, using in vitro and in vivo models, we further explored the mechanism by which BUB1 regulates ferroptosis and contributes to the resistance of PC cells to GEM. Additionally, we preliminarily examined the therapeutic efficacy of combining BUB1 inhibition with GEM treatment in PC. By elucidating the role of BUB1 in regulating ferroptosis and GEM resistance, our study provides valuable insights into the development of malignant tumors such as PC and offers potential clues for the development of novel therapeutic strategies.

## 2. Materials and Methods

### 2.1. Clinical Sample Collection

To obtain RNA-seq data and clinical features of 179 PC (T-group) samples and 171 normal samples from The Cancer Genome Atlas (TCGA) database (https://tcga-data.nci.nih.gov/tcga/, accessed on 6 March 2022), and to investigate the expression of BUB1 in normal and PC tissues. In this study, Transcripts Per Million (TPM) was used as the measure of gene expression. Box plots were generated from the database to compare the differences in BUB1 expression levels between PC patients and individuals without pancreatic-related diseases. Subsequently, based on the gene expression levels of BUB1, the PC patients were divided into two groups, each consisting of 89 cases. The survival curves of these patients were plotted with the time of onset as the zero point on the x-axis to evaluate the overall average lifespan. Additionally, the 179 PC patients were grouped according to PC staging, and violin plots were generated to observe the expression levels of BUB1 in different stages of PC.

### 2.2. Cell Culture and Treatment

PANC-1 (RRID:CVCL_0480), PATU-8988T (RRID:CVCL_1847), CFPAC-1 (RRID:CVCL_1119), BXPC-3 (RRID:CRL-1687), hTERT-HPNE(RRID:CVCL_C466), MIA Paca-2 (RRID:CRL-1420), and PANC-1/GEM (PANC-1 cells that are resistant to gemcitabine) cells were cultured in DMEM medium (Invitrogen, Carlsbad, CA, USA) containing 10% fetal bovine serum (Sigma, St. Louis, MO, USA). All cells were cultured at 37 °C with 5% CO_2_.

Construction of BUB1 knockdown stable cell lines: Log-phase growing PANC-1 or MIA Paca-2 cells were harvested and resuspended in cell suspension, then seeded into culture dishes. After the cells adhered to the dish, a mixture of siRNA duplexes and transfection reagent (Invitrogen) was added to the culture medium containing cells. The cells were further incubated at 37 °C, 5% CO_2_, and saturated humidity for 12 h. After that, the cells were collected, and the transfection efficiency was assessed using Western blot analysis. The most efficient siRNA sequence targeting BUB1 was selected to construct BUB1-targeting shRNA using lentivirus packaging and transfection into PANC-1 and MIA Paca-2 cells. The cells were then continuously cultured to obtain BUB1 knockdown stable cell lines (siRNA-A: TACAACAGTGACCTCCATCAA; siRNA-B: CCTGGGTCAGAGTATAGATAT; siRNA-C: ACCAGTGAGTTCCTATCCAAA).

Transfection of BUB1 plasmid for BUB1 knockdown: Log-phase growing PANC-1/GEM cells were harvested and resuspended in cell suspension, then seeded into a 6-well plate. After the cells adhered to the plate, a mixture of BUB1 plasmid (siRNA-A: TACAACAGTGACCTCCATCAA) and transfection reagent (Invitrogen) was added to the wells of the 6-well plate. The cells were incubated for approximately 4 h, and the cell status was observed. If the cell status was acceptable, the cells were further incubated for 2 h and then the original culture medium was replaced with fresh culture medium. If the cell status was poor, the culture medium was immediately discarded and replaced with complete culture medium, and the cells were further incubated for 24 h.

### 2.3. Animals Models

Twenty-four 4-week-old male Balb/c nude mice were randomly divided into the following groups: PANC-1 group, PANC-1 + GEM group, PANC-1 + BUB1-KD group, and PANC-1 + BUB1-KD + Gem group. In the PANC-1 group and PANC-1 + GEM group, mice were subcutaneously injected with 100 μL of PANC-1 cell suspension at a density of 1–5 × 10^7^ cells/mL into the flank. In the PANC-1 + BUB1-KD group and PANC-1 + BUB1-KD + GEM group, mice were injected with the same volume and density of BUB1 knockdown PANC-1 cell suspension. Once the tumor volume reached 50–100 mm^3^, mice in the PANC-1 + GEM group and PANC-1 + BUB1-KD + GEM group were intraperitoneally injected with 20 mg/kg of GEM (LY 188011, MCE) every 3 days. During the treatment period, mice body weight and tumor size were measured every 3 days. The experiment was terminated when the tumor volume approached 2000 mm^3^ or when the longest tumor diameter reached approximately 20 mm. After euthanizing the mice, tumor tissues were harvested, weighed, and photographed. The tumor tissues were then divided in half, with one half fixed in polyformaldehyde for further analysis and the other half snap-frozen in liquid nitrogen and stored at −80 °C. At the end of the experiment, the mouse carcasses were disposed of in a harmless manner.

### 2.4. Quantitative Real-Time PCR (qPCR)

First, cells were lysed using Trizol. Then, chloroform, isopropanol, 70% ethanol, and DEPC water were sequentially added to extract and dissolve RNA samples. The concentration of RNA was determined. PCR reaction mixtures were prepared, and the cDNA samples were diluted 10-fold and used as templates for PCR analysis. The prepared samples were loaded into an ABI StepOne Plus real-time PCR machine for the reaction. After completion of the reaction, data were collected and analyzed.

### 2.5. IC50 Detection

PANC-1, BXPC-3, and MIA Paca-2 cell suspensions were separately seeded into a 96-well plate. The cells were treated with Erastin at a concentration of (0, 1.25, 2.5, 5, 10, 20, 40) μM for 24 h. After removing the culture medium, 100 μL of 10% Cell Counting Kit-8 (CCK8, Beyotime, Shanghai, China) solution was added to each well, and the optical density was measured at 450 nm to calculate cell viability. Based on the cell viability data, a curve was plotted to depict the relationship between drug concentration and cell viability. By fitting the curve, the inhibitory effect of the drug on the cells was determined, and the IC50 value was calculated.

### 2.6. Western Blot

Total protein was extracted from tissues and cells using lysis buffer, and the protein samples were denatured by boiling. Prepared protein samples were subjected to SDS-PAGE for electrophoretic separation until the target protein bands were well resolved. Subsequently, the proteins were transferred from the gel onto a PVDF membrane. The membrane was incubated in 5% non-fat milk at room temperature for 2 h to block non-specific binding. After blocking, the membrane was incubated with the corresponding primary antibodies overnight at 4 °C. The membrane was washed three times with TBST solution, followed by incubation with the appropriate secondary antibodies at room temperature for 1 h. The membrane was washed three times with TBST solution, and an ECL substrate was applied to the PVDF membrane for visualization. The antibodies used in this study were BUB1 (Abcam, Cambridge, UK, ab195268), Glutathione Peroxidase 4 (GPX4) (Abcam, ab125066), Solute Carrier Family 7 Member 11 (SLC7A11) (Cell Signaling Technology, Danvers, MA, USA, #12691), β-actin (Cell Signaling Technology, #3700), NF2 (Cell Signaling Technology, #12888), MOB1 (Cell Signaling Technology, #3863), YAP (Cell Signaling Technology, #14074), and GAPDH (Boster, Pleasanton, CA, USA, A00227-1).

### 2.7. Real-Time Cell Analysis (RTCA)

Log-phase growing cells were seeded into E-Plate L8 wells. After the cells adhered to the plate, they were treated with or without 10 μM Erastin. The plate was then placed in an RTCA detection instrument (Agilent, Santa Clara, CA, USA) and cultured at 37 °C and 5% CO_2_. After 24 h of incubation, the detection was stopped, and the data were recorded.

### 2.8. Cell Cloning Experiment

Log-phase growing PANC-1 or MIA Paca-2 cells were harvested to create cell suspensions. The cell suspensions were diluted to an appropriate concentration and seeded into culture dishes. The dishes were then placed in a cell culture incubator at 37 °C, 5% CO_2_, and saturated humidity. After the cells adhered to the dish, they were treated with or without 10 μM Erastin for 24 h. The culture medium was replaced, and the cells were continued to be cultured. When visible clones appeared in the wells of a 6-well plate, the culture was terminated. The culture medium was discarded, and the cells were washed twice with PBS. The cells were fixed with 4% paraformaldehyde for 10 min and stained with 0.1% crystal violet solution. After washing the cells three times with PBS, the plate was air-dried. The clones were observed and counted under a microscope.

### 2.9. Transwell Assay

Before preparing the cell suspension, PANC-1 and MIA Paca-2 cells were subjected to serum starvation using serum-free basal medium. A cell suspension containing 3 × 10^5^ cells/mL was added to the Transwell chamber plate, with each well receiving 0.3 mL of cell suspension. Complete culture medium containing 10% FBS was added to the lower chamber of a 24-well plate, with three replicate wells for each group. The cells were treated with or without 10 μM Erastin and incubated at 37 °C for 24 h. Then, 1 mL of 4% paraformaldehyde solution was added to each well to fix the cells for 10 min at room temperature. The fixative was removed, and the cells were washed with PBS. Next, 1 mL of 0.5% crystal violet solution was added to each well and incubated for 30 min. The cells were washed three times with PBS, air-dried, and observed under a microscope in the Transwell chamber plate. Photographs were taken.

### 2.10. CCK8

PANC-1 cell suspensions were separately seeded into a 96-well plate. The cells were treated with GEM at concentrations of 0, 12.5, 25, 50, 100, 200, and 400 μM for 24 h. After removing the culture medium, 100 μL of 10% CCK8 solution was added to each well, and the optical density was measured at 450 nm to calculate cell viability.

PANC-1/GEM cell suspensions were seeded into a 96-well plate. The cells were treated with Erastin at concentrations of 0, 5, 10, and 20 μM for 24 h. After removing the culture medium, 100 μL of 10% CCK8 solution was added to each well, and the optical density was measured at 450 nm to calculate cell viability.

### 2.11. ROS Detection

According to the instructions of the ROS assay kit (S0033M, Beyotime, Shanghai, China), DCFH-DA was diluted 1:1000 with serum-free culture medium to obtain a concentration of 10 μM DCFH-DA solution. The cell culture medium was removed, and 1 mL of the diluted DCFH-DA solution was added to each well. The plate was then incubated at 37 °C in a cell culture incubator for 20 min. After incubation, the cells were washed three times with serum-free culture medium to remove any DCFH-DA that did not enter the cells. Subsequently, the samples were subjected to detection using a CytoFLEX flow cytometer (Beckman Coulter, Brea, CA, USA).

### 2.12. Immunofluorescence Technique

Tissue sections were prepared using a paraffin microtome to obtain 3–5 μm thick slices. The tissues were then sectioned and mounted on adhesive glass slides. After deparaffinization, antigen retrieval, and blocking, the samples were incubated overnight at 4 °C with Ki67 antibody (Cell Signaling Technology, 9449T). On the following day, the samples were exposed to SignalStain^®^ Boost Detection Reagent (Cell Signaling Technology, Danvers, MA, USA, HRP, Mouse #8125) at room temperature for 30 min, washed, and incubated with SignalStain^®^ DAB Substrate Kit (Cell Signaling Technology, Danvers, MA, USA, 8059). After sealing, the samples were examined under a microscope.

### 2.13. Data Analysis

All analyses were performed using the GraphPad Prism 8 Software. A one-way ANOVA or Student’s *t*-test was used to analyze statistical differences. All data are presented as the mean with SD from at least three individual experiments. *p* < 0.05 was considered statistically significant.

## 3. Results

### 3.1. High Expression of BUB1 in PC and its Association with Poor Prognosis

To investigate the relationship between BUB1 and PC, we utilized TGA-Assembler to study the expression levels of BUB1 in 179 PC patients (T-group) and 171 healthy individuals without pancreatic-related diseases (N-group) from The Cancer Genome Atlas (TCGA) database. The results revealed significantly higher expression of BUB1 in PC patients compared to healthy individuals (Figure 1A). Furthermore, PC patients with high BUB1 expression exhibited shorter overall mean survival and poorer prognosis (Figure 1B). Additionally, based on the staging of PC, we observed differential expression levels of BUB1 in different stages of PC tissues (Figure 1C), indicating a close association between BUB1 and PC. The expression level of BUB1 is relatively dispersed in PC I phase, while in PC II phase, the expression level of BUB1 is distributed unevenly. In PC III and PC IV phases, the expression level of BUB1 is relatively concentrated.

### 3.2. Erastin Downregulates the Expression of BUB1 and GPX4 While Upregulating the Expression of SLC7A11 in PC Cell Lines

In this study, the expression levels of BUB1 were detected using PCR in five PC cell lines (PANC-1, 8988, CFPAC, BXPC-3, MIA Paca-2) and normal PC line HPNE. The results, as shown in Figure 2A, revealed that PANC-1, BXPC-3, and MIA PaCa-2 cells exhibited the highest levels of BUB1 expression. We further determined the sensitivity of the selected three cell lines to Erastin, measuring the IC50 as an indicator of Erastin-induced apoptosis in pancreatic cancer cell lines. As depicted in Figure 2B–D, PANC-1 and MIA PaCa-2 cell lines exhibited lower IC50 values, indicating their higher sensitivity to Erastin. Consequently, we ultimately chose PANC-1 and MIA Paca-2 cell lines, which demonstrated high BUB1 expression and greater sensitivity to Erastin, for further experiments.

Further detection using PCR and Western blot was conducted to examine the expression levels of BUB1 and ferroptosis-related genes (GPX4, SLC7A11) in PC cells after treatment with Erastin. The results, as shown in Figure 2E–H, demonstrated significant downregulation of BUB1 expression and significant downregulation of GPX4 expression, along with a significant upregulation of SLC7A11 expression in the Erastin-treated group compared to the NC group. However, typically, the ferroptosis process is often accompanied by the depletion of GPX4 and SLC7A11 [26,27]. SLC7A11 is a component of the cystine/glutamate antiport system Xc-, responsible for the transport of cystine and glutamate into the cells. Upregulation of SLC7A11 usually indicates increased cystine uptake into the cells, leading to elevated intracellular cysteine levels and increased synthesis of glutathione, which is often accompanied by upregulation of the GSH-dependent enzyme GPX4 [27]. Sometimes cells induce an adaptive compensatory mechanism to protect cells from stress, including reactive oxygen species, by upregulating xCT expression and the function of system Xc-. This suggests that the upregulation of SLC7A11 may be compensatory, and its function is still suppressed, resulting in the decrease in GPX4 levels and induction of ferroptosis.

### 3.3. BUB1 Knockdown Enhances the Sensitivity of PC Cell Lines to Erastin, Inhibits Cell Proliferation and Migration

To further investigate the relationship between the BUB1 gene and ferroptosis in PC cells, we constructed a BUB1 knockdown model in PC cells and induced ferroptosis using Erastin. Based on the results shown in Figure 3A, we selected the A sequence siRNA with the highest interference efficiency to construct a shRNA lentivirus. PANC-1 and MIA Paca-2 cells were then transfected with the shRNA lentivirus to knock down the expression of BUB1. The successful establishment of PC cell lines with stable low expression of BUB1 is shown in Figure 3B.

Firstly, cell viability was assessed using RTCA in different cell groups. As shown in Figure 3C, under the influence of Erastin, PC cell lines exhibited a decline in cell viability, regardless of BUB1 knockdown. However, the reduction in cell viability was more pronounced in the shBUB1 group compared to the PC cell lines without BUB1 knockdown. Additionally, we observed that the cell viability of the shBUB1 group without Erastin treatment was lower than that of the NC group. Cell cloning experiments were performed to further investigate the impact of BUB1 on the proliferative capacity of PC cells during ferroptosis. It was observed that both Erastin treatment and shBUB1 treatment significantly inhibited cell proliferation (Figure 3D). Moreover, compared to Erastin treatment alone, the combination of Erastin and shBUB1 treatment led to a more pronounced reduction in cell numbers in PANC-1 and MIA Paca-2 cells. Transwell assays were conducted to evaluate the migratory ability of cells in different groups. As shown in Figure 3E, both Erastin treatment and shBUB1 treatment significantly inhibited the migration of PANC-1 and MIA Paca-2 cells. Furthermore, the combination of Erastin and shBUB1 treatment exhibited a more pronounced inhibitory effect on cell migration compared to Erastin treatment alone. These results indicate that low expression of BUB1 enhances the sensitivity of PC cells to Erastin, inhibits cell proliferation and migration, and suggests that BUB1 has a certain inhibitory effect on Erastin-induced ferroptosis in PC cells.

### 3.4. BUB1 Suppresses Ferroptosis in PC Cells by Modulating the NF2/MOB1-YAP Signaling Pathway, and BUB1 Knockdown Significantly Enhances the Sensitivity of Drug-Resistant PC Cells to Erastin

In Figure 4A, under low-dose GEM treatment, the impact of BUB1 knockdown on PANC-1 cell viability is more pronounced, indicating that BUB1 has a protective effect on PC cells against GEM toxicity. However, as the concentration of GEM increases, there is no significant difference in cell viability between the PANC-1 group and the PANC-1 + BUB1-KD group. This may be due to the excessively high concentration of GEM, resulting in a large number of cell deaths. Western blot results show that during the development of GEM resistance in PANC-1 cells, BUB1 expression is significantly upregulated, while the expression levels of NF2 and MOB1 proteins are downregulated, and the expression level of YAP protein is upregulated (Figure 4B). This suggests that BUB1, NF2, MOB1, and YAP are involved in regulating the resistance of PC cells to GEM.

PANC-1/GEM cells were divided into the following groups: PANC-1/GEM, PANC-1/GEM + plasmid, PANC-1/GEM + Erastin (10 μM), and PANC-1/GEM + plasmid + Erastin (10 μM), to further explore the mechanism of BUB1 resistance. CCK-8 results show that compared to the group treated with Erastin alone, the combination of BUB1 knockdown and Erastin treatment leads to a more significant decrease in PANC-1/GEM cell viability (Figure 4C). Additionally, we measured the ROS levels in each group and found that both BUB1 knockdown and Erastin treatment significantly upregulate ROS levels in PANC-1/GEM cells. Moreover, compared to Erastin treatment alone, the combination of BUB1 knockdown and Erastin treatment results in a more significant increase in ROS levels in PANC-1/GEM cells (Figure 4D). These results indicate that BUB1 promotes resistance of PC cells to GEM by inhibiting ferroptosis, and knockdown of BUB1 enhances the sensitivity of PANC-1/GEM cells to Erastin. Further investigation of the effect of BUB1 on the NF2/MOB1-YAP signaling pathway revealed that BUB1 knockdown and Erastin stimulation significantly upregulate the expression levels of NF2 and MOB1 proteins in PANC-1/GEM cells, while downregulating the expression levels of BUB1 and YAP proteins (Figure 4E). This suggests that both BUB1 and the NF2/MOB1-YAP signaling pathway are involved in Erastin-induced ferroptosis. Furthermore, compared to the group treated with Erastin alone, the combination of BUB1 knockdown and Erastin treatment leads to upregulation of NF2 and MOB1 protein expression levels and downregulation of YAP protein expression levels (Figure 4E). These results indicate that BUB1 inhibits ferroptosis in PC cells by modulating the NF2/MOB1-YAP signaling pathway, thereby promoting resistance of PC cells to GEM.

### 3.5. BUB1 Promotes GEM Resistance in PC Cells by Modulating the NF2/MOB1-YAP Signaling Pathway, and BUB1 Knockdown Significantly Enhances the Therapeutic Effect of GEM in PC

To further validate the mechanism by which BUB1 promotes GEM resistance in PC cells, a PANC-1 xenograft mouse model was established. As shown in Figure 5A, compared to the PANC-1 group, treatment with GEM or BUB1 knockdown significantly reduced tumor volume and weight in mice. Furthermore, when BUB1 knockdown was combined with GEM treatment, there was a further reduction in tumor volume and weight. Ki67 protein is highly overexpressed in cancer cells and is commonly used to assess tumor cell proliferation [28]. Immunofluorescence results showed that compared to the PANC-1 group, both GEM treatment and BUB1 knockdown led to a significant decrease in Ki67 staining in tumor tissues. Moreover, compared to GEM treatment alone, the combination of BUB1 knockdown and GEM treatment further reduced Ki67 staining in tumor tissues (Figure 5B). These findings indicate that BUB1 upregulates mouse PC cell proliferation, promotes mouse PC development, and plays a protective role in mouse PC cells during GEM administration. Additionally, inhibiting BUB1 significantly enhances the therapeutic effect of GEM in PC.

The changes in the NF2/MOB1-YAP signaling pathway in tumor cells after BUB1 knockdown and GEM treatment were examined using Western blot (Figure 5C). The results showed that both GEM treatment and BUB1 knockdown upregulated the expression levels of NF2 and MOB1 proteins, while downregulating the expression levels of YAP protein. Furthermore, compared to GEM treatment alone, the combination of BUB1 knockdown and GEM treatment further upregulated the expression levels of NF2 and MOB1 proteins and downregulated the expression levels of YAP protein. These findings suggest that BUB1 modulates the NF2/MOB1-YAP signaling pathway to promote PC cell proliferation and PC development. Knockdown of BUB1 significantly enhances the therapeutic effect of GEM in PC.

## 4. Discussion

Traditional treatments for PC include surgery, chemotherapy, radiation therapy, and palliative care, and the specific treatment strategy often depends on the stage of PC. Complete surgical resection significantly prolongs patient survival and is considered the only curative method for PC [1]. However, due to the nonspecific symptoms and gradual progression of PC, only about 15% of PC patients are suitable candidates for surgery [29]. Combination therapy with GEM and other chemotherapy drugs and adjuvant chemotherapy remain the preferred treatment options for advanced PC [30]. Patients show improvement in their condition during the early stages of chemotherapy; however, most patients develop chemotherapy drug resistance later, severely affecting their prognosis [31,32,33,34]. Understanding the mechanisms underlying GEM resistance in chemotherapy becomes crucial. Based on this, we conducted research at clinical, cellular, and animal levels and found that the molecule BUB1, which is highly expressed in PC patients, may play a significant role in PC resistance to GEM. Inhibiting BUB1 may have a beneficial effect in overcoming GEM chemotherapy resistance.

It is well known that BUB1 is involved in cell cycle regulation and the maintenance of chromosomal stability. Piao et al. found that BUB1 is significantly overexpressed in PC patients, and its high expression is associated with reduced overall survival in PC [16,21], which is consistent with our research findings. BUB1 can serve as a biomarker for predicting the prognosis of PC patients [21], with higher expression of BUB1 being associated with poorer prognosis. In addition, in our study, we observed no significant differences in BUB1 expression levels among different stages. Although BUB1 expression has an impact on prognosis, cancer prognosis is influenced by multiple factors, including tumor stage, histological grade, genetic alterations, and molecular markers. While staging is an important prognostic factor widely used in clinical practice, it is not the sole determinant of patient prognosis. Furthermore, tumors exhibit heterogeneity, and pancreatic cancer is particularly known for its heterogeneity at both genetic and phenotypic levels [35,36]. This heterogeneity can result in variations in BUB1 expression within tumors and among different patients. The staging system primarily considers anatomical features and tumor size, which may not fully capture molecular heterogeneity and its impact on prognosis. Therefore, BUB1 expression can serve as an additional prognostic marker complementing staging but may not align completely with staging. Zhang et al. identified the upregulation of BUB1 as an important factor in GEM resistance in bladder cancer cells [22]. Similarly, in PC cells, we observed significant upregulation of BUB1 in drug-resistant cells, and knocking down BUB1 could reverse the resistance to GEM. Therefore, the high expression of BUB1 may be an important factor in PC cell resistance to GEM. Additionally, we observed that BUB1 gene knockdown inhibited the growth and metastatic ability of PC cells and showed synergistic therapeutic effects when combined with GEM. Other related studies also support similar findings. For example, a study found that high expression of BUB1 in PC is associated with resistance to the chemotherapeutic drug 5-fluorouracil [25]. Furthermore, BUB1 gene knockout enhances the cytotoxicity of 5-FU in tumor cells [25]. This is consistent with our research findings and further supports the importance of BUB1 as a regulator of drug resistance.

Ferroptosis is an iron-dependent, programmed cell death process. Extensive research has demonstrated a close association between ferroptosis and GEM resistance in various tumor types, including PDAC [4,8,37]. Although there is currently no research linking BUB1 to ferroptosis, our study found that treatment of PC cells with the ferroptosis inducer Erastin significantly downregulates the expression of BUB1. Furthermore, low expression of BUB1 enhances the sensitivity of PC cells to Erastin, inhibiting cell proliferation, invasion, and migration. Studies have shown that YAP is significantly overexpressed in PDAC, and active YAP promotes cell proliferation and metastasis in PC [38,39]. Moreover, YAP has been identified as an independent prognostic marker for PDAC, with increased expression being associated with poor prognosis [40,41,42]. YAP is a core effector of the Hippo signaling pathway, and its expression is regulated by Hippo pathway inhibitory molecules such as NF2 and MOB1 [11,12,13,14]. In this study, we found that the expression levels of NF2 and MOB1 were downregulated, while the expression level of YAP was upregulated in drug-resistant PC cells. However, knocking down BUB1 or treating cells with a ferroptosis inducer significantly reversed these effects. Therefore, BUB1 can inhibit ferroptosis and promote GEM resistance in PC cells by downregulating the expression levels of NF2 and MOB1 and upregulating YAP expression. Similar findings have been reported in other types of cancer. For example, in sorafenib-resistant liver cancer cells, ferroptosis can be induced by downregulating YAP/TAZ [43]. Additionally, we found that inhibiting BUB1 further enhances the sensitivity of PC cells to Erastin-induced ferroptosis, providing a reasonable explanation for the synergistic therapeutic effect of GEM and BUB1 inhibition.

## 5. Conclusions

In conclusion, our research findings suggest that BUB1 promotes GEM resistance in PC cells by inhibiting the expression levels of NF2 and MOB1, leading to increased expression of YAP and suppression of ferroptosis. Additionally, the combination of GEM and BUB1 inhibition significantly enhances the cytotoxic effect on PC cells, displaying a synergistic therapeutic effect. These insights provide important implications for the development of strategies targeting GEM resistance, and the combination of BUB1 inhibitors may be an effective approach to overcome GEM resistance in PC chemotherapy. However, further research is needed to elucidate the underlying mechanisms of BUB1 in PC drug resistance and to validate its potential value in clinical treatment.

## Figures and Tables

**Figure 1 cancers-16-01540-f001:**
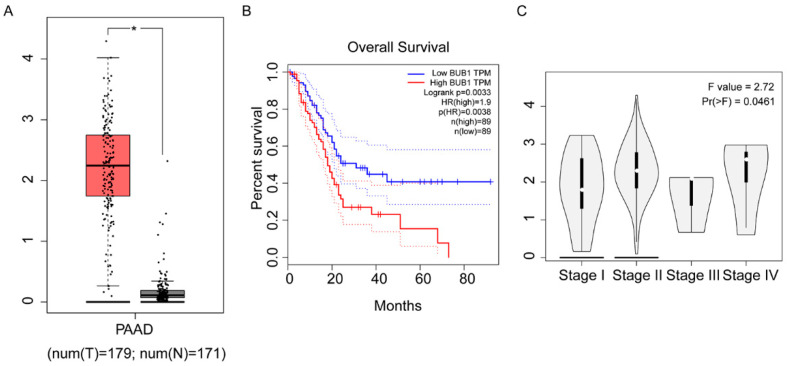
High expression of BUB1 in PC and its association with poor prognosis. (**A**) Expression levels of BUB1 in healthy individuals and PC patients; (**B**) survival curve of PC patients; (**C**) expression levels of BUB1 in PC at different stages. TPM: Transcripts Per Million.

**Figure 2 cancers-16-01540-f002:**
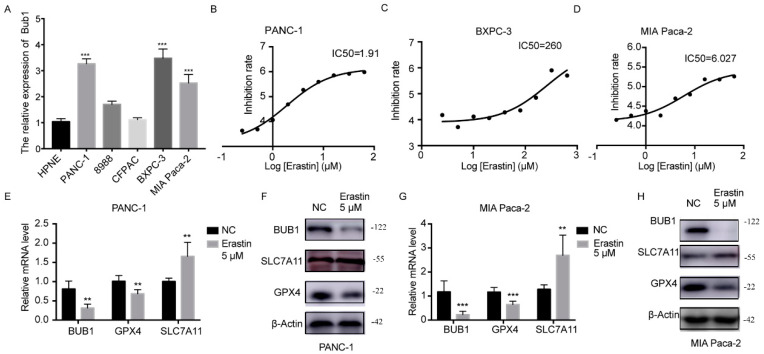
Erastin downregulates the expression of BUB1 and GPX4 while upregulating the expression of SLC7A11 in PC cell lines. (**A**) mRNA expression levels of BUB1 in different PC cell lines and healthy pancreatic cells detected using PCR, compared with HPNE, *** *p* < 0.001; (**B**−**D**) IC50 values of different PC cell lines determined using CCK8 assay; (**E**,**F**) mRNA and protein expression levels of BUB1, GPX4, and SLC7A11 in PANC−1 cells detected using PCR and Western blot, compared with NC, ** *p* < 0.01; (**G**,**H**) mRNA and protein expression levels of BUB1, GPX4, and SLC7A11 in MIA Paca-2 cells detected using PCR and Western blot, compared with NC, ** *p* < 0.01, *** *p* < 0.001. The uncropped blots are shown in Appendix A.

**Figure 3 cancers-16-01540-f003:**
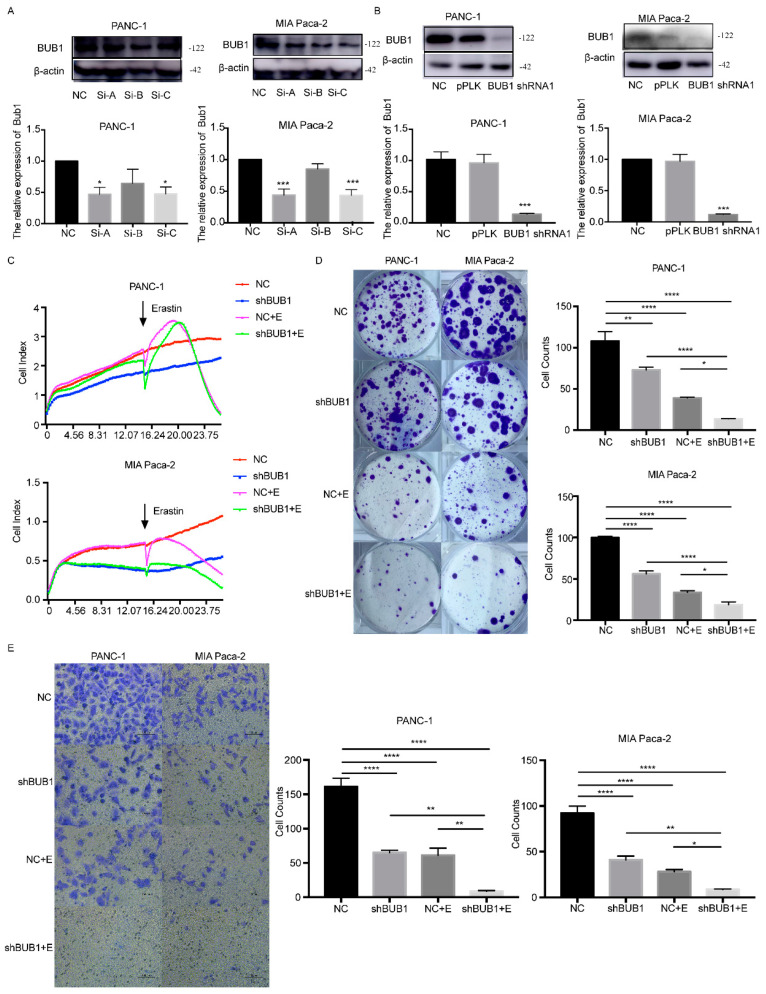
BUB1 knockdown enhances the sensitivity of PC cell lines to Erastin, inhibits cell proliferation, and migration. (**A**,**B**) Western blot analysis of BUB1 expression levels in PANC-1 and MIA Paca cells; (**C**) cell viability assessed by RTCA in PANC-1 and MIA Paca cells; (**D**) cell cloning experiments measuring the proliferative capacity of PANC-1 and MIA Paca cells; (**E**) transwell assay evaluating the migratory ability of PANC-1 and MIA Paca cells. Compared with NC (scale bar: 100 µm), * *p* < 0.05, ** *p* < 0.01, *** *p* < 0.001, **** *p* < 0.0001. The uncropped blots are shown in Appendix A.

**Figure 4 cancers-16-01540-f004:**
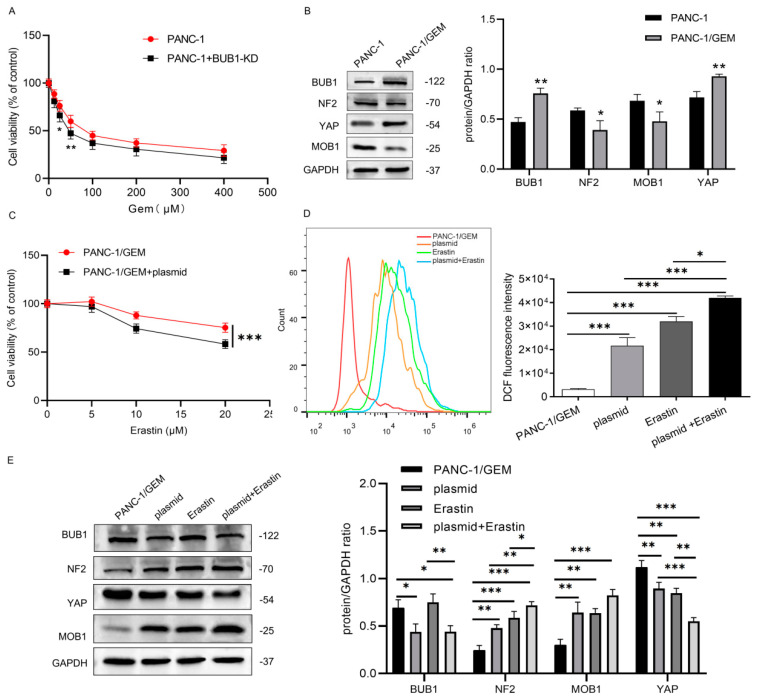
BUB1 suppresses ferroptosis in PC cells by modulating the NF2/MOB1−YAP signaling pathway, and BUB1 knockdown significantly enhances the sensitivity of drug-resistant PC cells to Erastin. (**A**) CCK−8 assay measuring cell viability in PANC−1 cells; (**B**) Western blot analysis of BUB1, NF2, MOB1, and YAP protein expression levels in PANC−1 cells and PANC−1/GEM cells, compared with PANC−1, * *p* < 0.05, ** *p* < 0.01; (**C**) CCK−8 assay assessing cell viability in PANC-1/GEM cells; (**D**) flow cytometry analysis of ROS levels in PANC−1/GEM cells, compared with PANC−1/GEM, * *p* < 0.05, *** *p* < 0.001; (**E**) Western blot analysis of BUB1, NF2, MOB1, and YAP protein expression levels in PANC−1/GEM cells, compared with PANC−1/GEM, * *p* < 0.05, ** *p* < 0.01, *** *p* < 0.001. The uncropped blots are shown in Appendix A.

**Figure 5 cancers-16-01540-f005:**
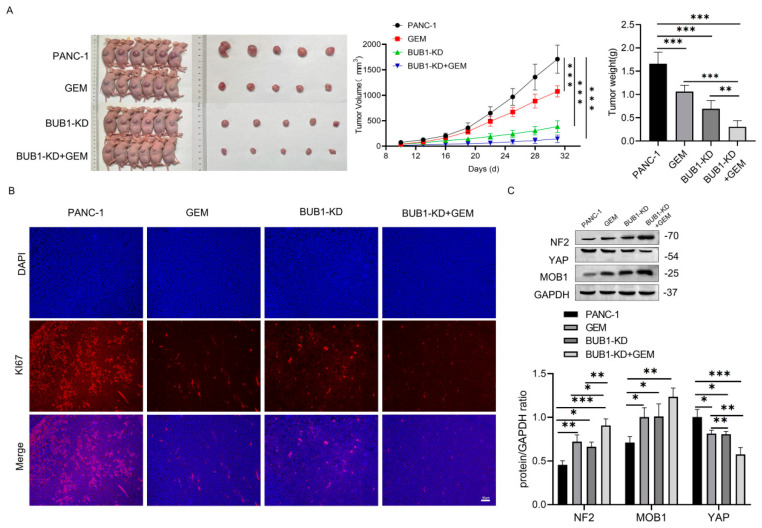
BUB1 promotes GEM resistance in PC cells by modulating the NF2/MOB1−YAP signaling pathway, and BUB1 knockdown significantly enhances the therapeutic effect of GEM in PC. (**A**) Tumor growth curve and volume in nude mice; (**B**) immunofluorescence detection of Ki67 staining in tumor tissues of nude mice (scale bar: 50 µm); (**C**) Western blot analysis of NF2, MOB1, and YAP protein expression levels in tumor tissues of nude mice. Compared with PANC−1, * *p* < 0.05, ** *p* < 0.01, *** *p* < 0.001. The uncropped blots are shown in Appendix A.

## Data Availability

The data used to support the findings of this study are available from the corresponding author upon request.

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
