# Peer review of "BUB1 Promotes Gemcitabine Resistance in Pancreatic Cancer Cells by Inhibiting Ferroptosis"

_cancers, 2024, doi:10.3390/cancers16081540_

Round 1
Reviewer 1 Report
Comments and Suggestions for Authors
The manuscript "BUB1 Promotes Gemcitabine Resistance in Pancreatic Cancer 2 Cells by Inhibiting Ferroptosis" by Wang et. al., is a nice paper that explores the role of BUB1 in pancreatic cancer in cultured cells and mouse model. The findings are interesting and well presented. Data presented nicely supports the conclusions.
I have one minor concern: the western blot images in Figs 2, 3, 4 and 5 are very tightly cropped. The images should show sufficient areas all around the band and not cropped so tightly. Please crop the images to show the bands and adjoining background all around the bands.
Author Response
Thank you for your valuable suggestions, the above issues have been modified.
Uncropped images of the WB blot has been placed in a PDF file and uploaded as an attachment.

Reviewer 2 Report
Comments and Suggestions for Authors
The authors reported that ferroptosis is involved in GEM resistance, and the inhibition of BUB1 contributes to the improvement of this resistance.
1.Please assign appropriate figure numbers. At Fig 2, it is noted as Fig 1B. Additionally, there are some sections where numbers have not been assigned.
2. If the expression of BUB1 impacts prognosis, then it is worth discussing why staging and expression do not align in their discussion.
3. EMT is also reported to be deeply involved in GEM resistance. I would like the authors to discuss the relationship between Bub1, ferroptosis, and EMT.
4. Please provide the gel images without splitting the results of the Western blot.
5. Please discuss the reason why there is almost no difference in cell viability when knocking down BUB1 in Figure 4A, while a significant difference is observed in the mouse model.
6. It seems that there is a discrepancy between the differences observed in mRNA expression in Fig 2E and F and the differences observed in Western blot (WB) results. Would this indicate a difference in the ratio of mRNA expression to protein translation? Please explain in their disucussion.
7. Reviewer also noticed that there is almost no difference in cell viability between the NC and shBUB1 groups in Figure 3B. Please provide an explanation for this result as well.
Comments on the Quality of English LanguagePlease provide a certificate of proofreading from a translation agency.
Author Response
The authors reported that ferroptosis is involved in GEM resistance, and the inhibition of BUB1 contributes to the improvement of this resistance.
1.Please assign appropriate figure numbers. At Fig 2, it is noted as Fig 1B. Additionally, there are some sections where numbers have not been assigned.
Response: Thank you for your valuable feedback and suggestions. We have made modifications to Figure 2 to ensure that the results correspond to the image.
- If the expression of BUB1 impacts prognosis, then it is worth discussing why staging and expression do not align in their discussion.
Response: Thank you for your valuable feedback and suggestions. While BUB1 expression has an impact on prognosis, cancer prognosis is influenced by multiple factors, including tumor stage, histological grade, genetic alterations, and molecular markers. Although staging is an important prognostic factor widely used in clinical practice, it is not the sole determinant of patient prognosis. Other factors, such as BUB1 expression, may provide additional prognostic information that cannot be fully captured by staging alone. Furthermore, tumors exhibit heterogeneity, and pancreatic cancer is particularly known for its heterogeneity at both genetic and phenotypic levels [1-2]. This heterogeneity can result in variations in BUB1 expression within tumors and among different patients. The staging system primarily considers anatomical features and tumor size, which may not fully capture molecular heterogeneity and its impact on prognosis. Therefore, BUB1 expression can serve as an additional prognostic marker complementing staging but may not be fully consistent with staging.
- Bärthel S, Falcomatà C, Rad R, Theis FJ, Saur D. Single-cell profiling to explore pancreatic cancer heterogeneity, plasticity and response to therapy. Nat Cancer. 2023 Apr;4(4):454-467. doi: 10.1038/s43018-023-00526-x
- Connor AA, Gallinger S. Pancreatic cancer evolution and heterogeneity: integrating omics and clinical data. Nat Rev Cancer. 2022 Mar;22(3):131-142. doi: 10.1038/s41568-021-00418-1
- EMT is also reported to be deeply involved in GEM resistance. I would like the authors to discuss the relationship between Bub1, ferroptosis, and EMT.
Response: Thank you for your valuable feedback and suggestions. Thank you for your feedback and suggestions. You have provided us with a very valuable research direction. Previous studies have shown that knocking down BUB1 can suppress epithelial-mesenchymal transition (EMT), migration, and invasion in lung cancer cells [1]. It has also been observed that mesenchymal cancer cells are more susceptible to ferroptosis compared to their epithelial counterparts. Drug-resistant cancer cells undergoing EMT are more easily killed by inducers of ferroptosis [2]. In this study, we found that BUB1 promotes gemcitabine resistance in pancreatic cancer cells by inhibiting ferroptosis. However, since our study did not involve EMT and there is currently no research explaining the relationship between BUB1, ferroptosis, and EMT, this discussion was not included in the manuscript. Nevertheless, this is still a topic worth exploring, and we will consider investigating the relationship between BUB1, ferroptosis, and EMT in our future research.
[1]Nyati S, Schinske-Sebolt K, Pitchiaya S, Chekhovskiy K, Chator A, Chaudhry N, Dosch J, Van Dort ME, Varambally S, Kumar-Sinha C, Nyati MK, Ray D, Walter NG, Yu H, Ross BD, Rehemtulla A. The kinase activity of the Ser/Thr kinase BUB1 promotes TGF-β signaling. Sci Signal. 2015 Jan 6;8(358):ra1. doi: 10.1126/scisignal.2005379。
[2]Ren Y, Mao X, Xu H, Dang Q, Weng S, Zhang Y, Chen S, Liu S, Ba Y, Zhou Z, Han X, Liu Z, Zhang G. Ferroptosis and EMT: key targets for combating cancer progression and therapy resistance. Cell Mol Life Sci. 2023 Aug 19;80(9):263. doi: 10.1007/s00018-023-04907-4
- Please provide the gel images without splitting the results of the Western blot.
Response: Uncropped images of the WB blot has been placed in a PDF file and uploaded as an attachment.
- Please discuss the reason why there is almost no difference in cell viability when knocking down BUB1 in Figure 4A, while a significant difference is observed in the mouse model.
Response: Thank you for your valuable feedback and suggestions. According to our statistical results, under the treatment of 25 μM and 50 μM gemcitabine, knocking down BUB1 significantly inhibited cell viability compared to the control group. However, as the concentration of gemcitabine increased, this difference gradually decreased. This may be due to the high dose of gemcitabine causing extensive cell death, thereby masking the impact of BUB1 knockout on cells. Furthermore, the observed results in vitro were not as significant as the in vivo results, which could be attributed to the differences in reactions between in vitro and in vivo conditions, as well as tumor heterogeneity. Cell culture experiments provide a simplified and controlled environment that may not fully recapitulate the complex interactions and microenvironment present in actual organisms. On the other hand, in vivo mouse models involve multiple cellular and systemic factors that can influence the response to BUB1 knockdown. Pancreatic cancer exhibits significant heterogeneity within individual tumors and among different patients. This heterogeneity can manifest as variations in genetic alterations, cellular phenotypes, and molecular pathways. The specific cell subpopulation used in in vitro experiments may not fully represent the entire tumor population, leading to different responses compared to the mouse model. Additionally, differences in drug metabolism and pharmacokinetics may also contribute to the disparities between in vitro and in vivo results.
- It seems that there is a discrepancy between the differences observed in mRNA expression in Fig 2E and F and the differences observed in Western blot (WB) results. Would this indicate a difference in the ratio of mRNA expression to protein translation? Please explain in their disucussion.
Response: Thank you for your valuable feedback and suggestions. The mRNA expression differences in Figures 2E and F are largely consistent with the trend of Western blot. It is not known whether the order of arrangement is the reason for the misunderstanding because CPX4 and SLC7A11 are in a different order in the two figures.
- Reviewer also noticed that there is almost no difference in cell viability between the NC and shBUB1 groups in Figure 3B. Please provide an explanation for this result as well.
Response: The picture of cell viability mentioned by the reviewer should refer to 3C, which shows the difference in cell viability between the NC group and the shBUB1 group. We used Real-Time Cell Analysis (RTCA) to monitor the changes in cell viability. Our main objective was to observe the trends in cell viability. However, since the trends in cell viability are generated directly on the software, we are unable to perform statistical analysis between different groups.

Reviewer 3 Report
Comments and Suggestions for Authors
Dear authors, I would like to congratulate you on the successful conduct of this research, please see the following recommendations for minor changes:
For “BUB1, GPX-4 and SLC7A11, upon first occurrence in text, please specify what the abbreviation stands for. For example: Glutathione peroxidase 4 (GPX4)
Page 2. 2.2 cell culture, line 96: Please define with PANC-1/GEM cells mean? Are these PANC-1 cells that are resistant to gemcitabine? Please specify.
Section 2.6, line 163: “Cell Signaling Technology, #3700”: please correct the font size of #3700. This font seems larger that other numbers in this section.
For figures: please specify what level of significance *, **. ***, **** signifies.
Figure 1B:
please specify the number at risk for each group at each timeline.
Please specify what “TPM” stands for (define the abbreviation)
Figure 1C: Please specify which staging system you used, and how many patients belonged to each stage.
Section 3.2, line 247, “Figure 1A”: Please changed to “Figure 2A”
Section 3.3, line 271: “Figure A”->are you referring to Figure 3A? please specify.
Section 3.3, line 275: Please change “Figure 2B” to “Figure 3B”
Author Response
Dear authors, I would like to congratulate you on the successful conduct of this research, please see the following recommendations for minor changes:
For “BUB1, GPX-4 and SLC7A11, upon first occurrence in text, please specify what the abbreviation stands for. For example: Glutathione peroxidase 4 (GPX4)
Response: Thank you for your valuable feedback and suggestions. We have added the full names of "BUB1, GPX-4, and SLC7A11" in the manuscript.
Page 2. 2.2 cell culture, line 96: Please define with PANC-1/GEM cells mean? Are these PANC-1 cells that are resistant to gemcitabine? Please specify.
Response: Thank you for your valuable feedback and suggestions. PANC-1/GEM cells are the gemcitabine-resistant cell line we purchased.
Section 2.6, line 163: “Cell Signaling Technology, #3700”: please correct the font size of #3700. This font seems larger that other numbers in this section.
Response: Thank you for your valuable feedback and suggestions. We have made the font modification for "#3700" to ensure consistency throughout the entire text.
For figures: please specify what level of significance *, **. ***, **** signifies.
Response: Thank you for your valuable feedback and suggestions. We have added explanations for the definitions of " *, **, ***, ****" in the annotations.
Figure 1B:
please specify the number at risk for each group at each timeline.
Please specify what “TPM” stands for (define the abbreviation)
Response: Since the results were generated from an analysis in the TCGA database, the number of patients per stage does not seem to be available.
TPM stands for "Transcripts Per Million," and we have added this information at the corresponding position in the manuscript.
Figure 1C: Please specify which staging system you used, and how many patients belonged to each stage.
Response: TNM staging system was utilized. Since the results were generated from an analysis in the TCGA database, the number of patients per stage does not seem to be available.
Section 3.2, line 247, “Figure 1A”: Please changed to “Figure 2A”
Response: Thank you for your valuable feedback and suggestions. We have made the necessary modifications at the respective location in the manuscript.
Section 3.3, line 271: “Figure A”->are you referring to Figure 3A? please specify.
Response: Thank you for your valuable feedback and suggestions. "Figure A" in the manuscript refers to Figure 3A. We apologize for the oversight that caused confusion, and we have reviewed the entire manuscript and made the necessary corrections.
Section 3.3, line 275: Please change “Figure 2B” to “Figure 3B”
Response: Thank you for your valuable feedback and suggestions. Thank you for your valuable feedback and suggestions. We have made the necessary modifications at the respective location in the manuscript.
Reviewer 4 Report
Comments and Suggestions for Authors
General: This is a manuscript describing the role of BUB1 in the process of ferroptosis and the mechanisms by which BUB1 regulates ferroptosis and contributes to gemcitabine resistance. Indeed, with this study, the authors suggest that BUB1 promotes gemcitabine resistance in pancreatic cancer cells by inhibiting ferroptosis. Additionaly, the manuscript also explores the possible synergistic effect of combining gemcitabine with BUB1 inhibition.
Minor issues:
1. Please check spell and language flow along the manuscript.
2. line 28 is used a reference with almost 4 years to provide information about incidence of PC globally. The Global Cancer Statistics have already published the same information in 2022/2023. Please update this information.
3. Why the using of this specific mouse model (Bulb/c nude)? Is this the Balb/c nude model, with a spelling error? What advantages gives this model to the proposed experiments? Is there any other models more appropriated?
4. Please check very carefully all the mentions to the figures along the manuscript. Some are with the wrong numbers and others are without them. (e.g line 247 "figure 1A" should be replaced with figure 2A and in line 371 it appears only "figure A"). There are more errors than the provided examples.
5. If you can, please comment a little further than the explanation provided for the increased SLC7A11 expression (the compensatory mechanism - line 265).
6. In the figure 2, are the IC50 values correct? The IC50 of the BXPC3 is really 260? This value is in which measure unity, micromolar?
7. Still in figure 2, panel E and F, the relative mRNA level is normalized to what? To NC cells? If yes, why some of them are lower and higher than 1?
8. Still in figure 2, panel F, the representative blot of BUB1 does not exhibit the difference presented in the graph. Why?
9. Lastly, what are the PANC-1/GEM cells? Are PANC-1 cells treated with GEM? Or a PANC-1 cell line resistant to GEM? And how this resistance was acquired?
Author Response
Minor issues:
1. Please check spell and language flow along the manuscript.
Response: Thank you for your valuable feedback. It's already been changed in the revised manuscript.
- line 28 is used a reference with almost 4 years to provide information about incidence of PC globally. The Global Cancer Statistics have already published the same information in 2022/2023. Please update this information.
Response: Thank you for your valuable feedback. It's already been changed in the revised manuscript.
- Why the using of this specific mouse model (Bulb/c nude)? Is this the Balb/c nude model, with a spelling error? What advantages gives this model to the proposed experiments? Is there any other models more appropriated?
Response: Thank you for your valuable feedback. We are sorry for the spelling error. We used Balb/c nude mice. The biggest feature of the nude mice used in tumor formation experiments is that there is no immune organs such as thymus, its immunity is almost zero, it has no immune effect on tumor cells, and it is easy to tumor formation. C57BL/6 wild-type Mice can be used in situ modeling. We did not use it in this experiment, which required a laparotomy and in-situ implantation of cells after finding pancreatic tissue. It requires certain surgical skills.
- Please check very carefully all the mentions to the figures along the manuscript. Some are with the wrong numbers and others are without them. (e.g line 247 "figure 1A" should be replaced with figure 2A and in line 371 it appears only "figure A"). There are more errors than the provided examples.
Response: Thank you for your valuable suggestions, the above issues have been modified.
- If you can, please comment a little further than the explanation provided for the increased SLC7A11 expression (the compensatory mechanism - line 265).
Response: Thank you for your valuable suggestions. Changes have been made in the revised manuscript.
- In the figure 2, are the IC50 values correct? The IC50 of the BXPC3 is really 260? This value is in which measure unity, micromolar?
Response: Yes, the IC50 values is correct. In our experiment, BXPC3 cells is very insensitive to erastin. So, we did not use BXPC3 cells in subsequent experiments. The measure unity is Micromoles per liter.
- Still in figure 2, panel E and F, the relative mRNA level is normalized to what? To NC cells? If yes, why some of them are lower and higher than 1?
Response: The relative mRNA level is normalized to one of the NC cells. The experiment was repeated three times, with one of the NC cells as 1.
- Still in figure 2, panel F, the representative blot of BUB1 does not exhibit the difference presented in the graph. Why?
Response: Thank you for your valuable feedback and suggestions. The representative blot of BUB1 does not exhibit the difference presented in the gragh should refer to 2H. This may be because the WB picture we selected is not representative enough, we will change this picture.
- Lastly, what are the PANC-1/GEM cells? Are PANC-1 cells treated with GEM? Or a PANC-1 cell line resistant to GEM? And how this resistance was acquired?
Response: Thank you for your valuable feedback and suggestions. PANC-1/GEM cells are the gemcitabine-resistant cell line we purchased. The attachment is how the cells acquired.

Round 2
Reviewer 2 Report
Comments and Suggestions for Authors
Reviewer considered that the response appropriately addresses the reviewer's comments.